# ADVERSARIAL MACHINE LEARNING AT SCALE

**Alexey Kurakin**
Google Brain
kurakin@google.com

**Ian J. Goodfellow**
OpenAI
ian@openai.com

**Samy Bengio**
Google Brain
bengio@google.com

## ABSTRACT

Adversarial examples are malicious inputs designed to fool machine learning models. They often transfer from one model to another, allowing attackers to mount black box attacks without knowledge of the target model's parameters. Adversarial training is the process of explicitly training a model on adversarial examples, in order to make it more robust to attack or to reduce its test error on clean inputs. So far, adversarial training has primarily been applied to small problems. In this research, we apply adversarial training to ImageNet (Russakovsky et al., 2014). Our contributions include: (1) recommendations for how to successfully scale adversarial training to large models and datasets, (2) the observation that adversarial training confers robustness to single-step attack methods, (3) the finding that multi-step attack methods are somewhat less transferable than single-step attack methods, so single-step attacks are the best for mounting black-box attacks, and (4) resolution of a "label leaking" effect that causes adversarially trained models to perform better on adversarial examples than on clean examples, because the adversarial example construction process uses the true label and the model can learn to exploit regularities in the construction process.

## 1 INTRODUCTION

It has been shown that machine learning models are often vulnerable to adversarial manipulation of their input intended to cause incorrect classification (Dalvi et al., 2004). In particular, neural networks and many other categories of machine learning models are highly vulnerable to attacks based on small modifications of the input to the model at test time (Biggio et al., 2013; Szegedy et al., 2014; Goodfellow et al., 2014; Papernot et al., 2016b).

The problem can be summarized as follows. Let's say there is a machine learning system $M$ and input sample $C$ which we call a clean example. Let's assume that sample $C$ is correctly classified by the machine learning system, i.e. $M(C) = y_{true}$. It's possible to construct an adversarial example $A$ which is perceptually indistinguishable from $C$ but is classified incorrectly, i.e. $M(A) \neq y_{true}$. These adversarial examples are misclassified far more often than examples that have been perturbed by noise, even if the magnitude of the noise is much larger than the magnitude of the adversarial perturbation (Szegedy et al., 2014).

Adversarial examples pose potential security threats for practical machine learning applications. In particular, Szegedy et al. (2014) showed that an adversarial example that was designed to be misclassified by a model $M_1$ is often also misclassified by a model $M_2$. This adversarial example transferability property means that it is possible to generate adversarial examples and perform a misclassification attack on a machine learning system without access to the underlying model. Papernot et al. (2016a) and Papernot et al. (2016b) demonstrated such attacks in realistic scenarios.

It has been shown (Goodfellow et al., 2014; Huang et al., 2015) that injecting adversarial examples into the training set (also called adversarial training) could increase robustness of neural networks to adversarial examples. Another existing approach is to use defensive distillation to train the network (Papernot et al., 2015). However all prior work studies defense measures only on relatively small datasets like MNIST and CIFAR10. Some concurrent work studies attack mechanisms on ImageNet (Rozsa et al., 2016), focusing on the question of how well adversarial examples transfer between different types of models, while we focus on defenses and studying how well different types of adversarial example generation procedures transfer between relatively similar models.

In this paper we studied adversarial training of Inception models trained on ImageNet. The contributions of this paper are the following:

- We successfully used adversarial training to train an Inception v3 model (Szegedy et al., 2015) on ImageNet dataset (Russakovsky et al., 2014) and to significantly increase robustness against adversarial examples generated by the *fast gradient sign method* (Goodfellow et al., 2014) as well as other one-step methods.

- We demonstrated that different types of adversarial examples tend to have different transferability properties between models. In particular we observed that those adversarial examples which are harder to resist using adversarial training are less likely to be transferrable between models.

- We showed that models which have higher capacity (i.e. number of parameters) tend to be more robust to adversarial examples compared to lower capacity model of the same architecture. This provides additional cue which could help building more robust models.

- We also observed an interesting property we call "label leaking". Adversarial examples constructed with a single-step method making use of the true labels may be easier to classify than clean adversarial examples, because an adversarially trained model can learn to exploit regularities in the adversarial example construction process. This suggests using adversarial example construction processes that do not make use of the true label.

The rest of the paper is structured as follows: In section 2 we review different methods to generate adversarial examples. Section 3 describes details of our adversarial training algorithm. Finally, section 4 describes our experiments and results of adversarial training.

## 2 METHODS GENERATING ADVERSARIAL EXAMPLES

### 2.1 TERMINOLOGY AND NOTATION

In this paper we use the following notation and terminology regarding adversarial examples:

1. $\boldsymbol{X}$, the *clean image* — unmodified image from the dataset (either train or test set).

2. $\boldsymbol{X}^{adv}$, the *adversarial image*: the output of any procedure intended to produce an approximate worst-case modification of the clean image. We sometimes call this a *candidate adversarial image* to emphasize that an adversarial image is not necessarily misclassified by the neural network.

3. *Misclassified adversarial image* — candidate adversarial image which is misclassified by the neural network. In addition we are typically interested only in those misclassified adversarial images when the corresponding clean image is correctly classified.

4. $\epsilon$: The size of the adversarial perturbation. In most cases, we require the $L_\infty$ norm of the perturbation to be less than $\epsilon$, as done by Goodfellow et al. (2014). We always specify $\epsilon$ in terms of pixel values in the range $[0, 255]$. Note that some other work on adversarial examples minimizes the size of the perturbation rather than imposing a constraint on the size of the perturbation (Szegedy et al., 2014).

5. The cost function used to train the model is denoted $J(\boldsymbol{X}, y_{true})$.

6. $Clip_{\boldsymbol{X}, \epsilon}(\boldsymbol{A})$ denotes element-wise clipping $\boldsymbol{A}$, with $A_{i,j}$ clipped to the range $[X_{i,j} - \epsilon, X_{i,j} + \epsilon]$.

7. *One-step* methods of adversarial example generation generate a candidate adversarial image after computing only one gradient. They are often based on finding the optimal perturbation of a linear approximation of the cost or model. *Iterative* methods apply many gradient updates. They typically do not rely on any approximation of the model and typically produce more harmful adversarial examples when run for more iterations.

### 2.2 ATTACK METHODS

We study a variety of attack methods:

**Fast gradient sign method** Goodfellow et al. (2014) proposed the *fast gradient sign method* (FGSM) as a simple way to generate adversarial examples:

$$\boldsymbol{X}^{adv} = \boldsymbol{X} + \epsilon \, \text{sign}\big(\nabla_X J(\boldsymbol{X}, y_{true})\big) \tag{1}$$

This method is simple and computationally efficient compared to more complex methods like L-BFGS (Szegedy et al., 2014), however it usually has a lower success rate. On ImageNet, top-1 error rate on candidate adversarial images for the FGSM is about $63\% - 69\%$ for $\epsilon \in [2, 32]$.

**One-step target class methods** FGSM finds adversarial perturbations which increase the value of the loss function. An alternative approach is to maximize probability $p(y_{target} \mid \boldsymbol{X})$ of some specific target class $y_{target}$ which is unlikely to be the true class for a given image. For a neural network with cross-entropy loss this will lead to the following formula for the one-step target class method:

$$\boldsymbol{X}^{adv} = \boldsymbol{X} - \epsilon \, \text{sign}\big(\nabla_X J(\boldsymbol{X}, y_{target})\big) \tag{2}$$

As a target class we can use the least likely class predicted by the network $y_{LL} = \arg\min_y \{p(y \mid \boldsymbol{X})\}$, as suggested by Kurakin et al. (2016). In such case we refer to this method as one-step least likely class or just "step l.l." Alternatively we can use a random class as target class. In such a case we refer to this method as "step rnd.".

**Basic iterative method** A straightforward extension of FGSM is to apply it multiple times with small step size:

$$\boldsymbol{X}_0^{adv} = \boldsymbol{X}, \quad \boldsymbol{X}_{N+1}^{adv} = Clip_{X,\epsilon}\Big\{\boldsymbol{X}_N^{adv} + \alpha \, \text{sign}\big(\nabla_X J(\boldsymbol{X}_N^{adv}, y_{true})\big)\Big\}$$

In our experiments we used $\alpha = 1$, i.e. we changed the value of each pixel only by 1 on each step. We selected the number of iterations to be $\min(\epsilon + 4, 1.25\epsilon)$. See more information on this method in Kurakin et al. (2016). Below we refer to this method as "iter. basic" method.

**Iterative least-likely class method** By running multiple iterations of the "step l.l." method we can get adversarial examples which are misclassified in more than $99\%$ of the cases:

$$\boldsymbol{X}_0^{adv} = \boldsymbol{X}, \quad \boldsymbol{X}_{N+1}^{adv} = Clip_{X,\epsilon}\left\{\boldsymbol{X}_N^{adv} - \alpha \, \text{sign}\left(\nabla_X J(\boldsymbol{X}_N^{adv}, y_{LL})\right)\right\}$$

$\alpha$ and number of iterations were selected in the same way as for the basic iterative method. Below we refer to this method as the "iter. l.l.".

## 3 ADVERSARIAL TRAINING

The basic idea of adversarial training is to inject adversarial examples into the training set, continually generating new adversarial examples at every step of training (Goodfellow et al., 2014). Adversarial training was originally developed for small models that did not use batch normalization. To scale adversarial training to ImageNet, we recommend using batch normalization (Ioffe & Szegedy, 2015). To do so successfully, we found that it was important for examples to be grouped into batches containing both normal and adversarial examples before taking each training step, as described in algorithm 1.

We use a loss function that allows independent control of the number and relative weight of adversarial examples in each batch:

$$Loss = \frac{1}{(m-k) + \lambda k}\left(\sum_{i \in CLEAN} L(X_i|y_i) + \lambda \sum_{i \in ADV} L(X_i^{adv}|y_i)\right)$$

where $L(X|y)$ is a loss on a single example $X$ with true class $y$; $m$ is total number of training examples in the minibatch; $k$ is number of adversarial examples in the minibatch and $\lambda$ is a parameter which controls the relative weight of adversarial examples in the loss. We used $\lambda = 0.3$, $m = 32$,

---

**Algorithm 1** Adversarial training of network $N$.
Size of the training minibatch is $m$. Number of adversarial images in the minibatch is $k$.

---

1: Randomly initialize network $N$
2: **repeat**
3:　　Read minibatch $B = \{X^1, \ldots, X^m\}$ from training set
4:　　Generate $k$ adversarial examples $\{X^1_{adv}, \ldots, X^k_{adv}\}$ from corresponding
　　　clean examples $\{X^1, \ldots, X^k\}$ using current state of the network $N$
5:　　Make new minibatch $B' = \{X^1_{adv}, \ldots, X^k_{adv}, X^{k+1}, \ldots, X^m\}$
6:　　Do one training step of network $N$ using minibatch $B'$
7: **until** training converged

---

and $k = 16$. Note that we *replace* each clean example with its adversarial counterpart, for a total minibatch size of 32, which is a departure from previous approaches to adversarial training.

Fraction and weight of adversarial examples which we used in each minibatch differs from Huang et al. (2015) where authors replaced entire minibatch with adversarial examples. However their experiments was done on smaller datasets (MNIST and CIFAR-10) in which case adversarial training does not lead to decrease of accuracy on clean images. We found that our approach works better for ImageNet models (corresponding comparative experiments could be found in Appendix E).

We observed that if we fix $\epsilon$ during training then networks become robust only to that specific value of $\epsilon$. We therefore recommend choosing $\epsilon$ randomly, independently for each training example. In our experiments we achieved best results when magnitudes were drawn from a truncated normal distribution defined in interval $[0, 16]$ with underlying normal distribution $N(\mu = 0, \sigma = 8)$.[1]

## 4 EXPERIMENTS

We adversarially trained an Inception v3 model (Szegedy et al., 2015) on ImageNet. All experiments were done using synchronous distributed training on 50 machines, with a minibatch of 32 examples on each machine. We observed that the network tends to reach maximum accuracy at around $130k - 150k$ iterations. If we continue training beyond $150k$ iterations then eventually accuracy might decrease by a fraction of a percent. Thus we ran experiments for around $150k$ iterations and then used the obtained accuracy as the final result of the experiment.

Similar to Szegedy et al. (2015) we used RMSProp optimizer for training. We used a learning rate of 0.045 except where otherwise indicated.

We looked at interaction of adversarial training and other forms or regularization (dropout, label smoothing and weight decay). By default training of Inception v3 model uses all three of them. We noticed that disabling label smoothing and/or dropout leads to small decrease of accuracy on clean examples (by 0.1% - 0.5% for top 1 accuracy) and small increase of accuracy on adversarial examples (by 1% - 1.5% for top 1 accuracy). On the other hand reducing weight decay leads to decrease of accuracy on both clean and adversarial examples.

We experimented with delaying adversarial training by 0, $10k$, $20k$ and $40k$ iterations. In such case we used only clean examples during the first $N$ training iterations and after $N$ iterations included both clean and adversarial examples in the minibatch. We noticed that delaying adversarial training has almost no effect on accuracy on clean examples (difference in accuracy within 0.2%) after sufficient number of training iterations (more than $70k$ in our case). At the same time we noticed that larger delays of adversarial training might cause up to 4% decline of accuracy on adversarial examples with high magnitude of adversarial perturbations. For small $10k$ delay changes of accuracy was not statistically significant to recommend against it. We used a delay of $10k$ because this allowed us to reuse the same partially trained model as a starting point for many different experiments.

For evaluation we used the ImageNet validation set which contains $50,000$ images and does not intersect with the training set.

---

[1] In TensorFlow this could be achieved by `tf.abs(tf.truncated_normal(shape, mean=0, stddev=8))`.

## 4.1 RESULTS OF ADVERSARIAL TRAINING

We experimented with adversarial training using several types of one-step methods. We found that adversarial training using any type of one-step method increases robustness to all types of one-step adversarial examples that we tested. However there is still a gap between accuracy on clean and adversarial examples which could vary depending on the combination of methods used for training and evaluation.

Adversarial training caused a slight (less than $1\%$) decrease of accuracy on clean examples in our ImageNet experiments. This differs from results of adversarial training reported previously, where adversarial training increased accuracy on the test set (Goodfellow et al., 2014; Miyato et al., 2016b;a). One possible explanation is that adversarial training acts as a regularizer. For datasets with few labeled examples where overfitting is the primary concern, adversarial training reduces test error. For datasets like ImageNet where state-of-the-art models typically have high training set error, adding a regularizer like adversarial training can increase training set error more than it decreases the gap between training and test set error. Our results suggest that adversarial training should be employed in two scenarios:

1. When a model is overfitting, and a regularizer is required.
2. When security against adversarial examples is a concern. In this case, adversarial training is the method that provides the most security of any known defense, while losing only a small amount of accuracy.

By comparing different one-step methods for adversarial training we observed that the best results in terms or accuracy on test set are achieved using "step l.l." or "step rnd." method. Moreover using these two methods helped the model to become robust to adversarial examples generated by other one-step methods. Thus for final experiments we used "step l.l." adversarial method.

For brevity we omitted a detailed comparison of different one-step methods here, but the reader can find it in Appendix A.

Table 1: Top 1 and top 5 accuracies of an adversarially trained network on clean images and adversarial images with various test-time $\epsilon$. Both training and evaluation were done using "step l.l." method. Adversarially training caused the baseline model to become robust to adversarial examples but lost some accuracy on clean examples. We therefore also trained a deeper model with two additional Inception blocks. The deeper model benefits more from adversarial training in terms of robustness to adversarial perturbation, and loses less accuracy on clean examples than the smaller model does.

|  |  | Clean | $\epsilon = 2$ | $\epsilon = 4$ | $\epsilon = 8$ | $\epsilon = 16$ |
|---|---|---|---|---|---|---|
| Baseline | top 1 | 78.4% | 30.8% | 27.2% | 27.2% | 29.5% |
| (standard training) | top 5 | 94.0% | 60.0% | 55.6% | 55.1% | 57.2% |
| Adv. training | top 1 | 77.6% | 73.5% | 74.0% | 74.5% | 73.9% |
|  | top 5 | 93.8% | 91.7% | 91.9% | 92.0% | 91.4% |
| Deeper model | top 1 | 78.7% | 33.5% | 30.0% | 30.0% | 31.6% |
| (standard training) | top 5 | 94.4% | 63.3% | 58.9% | 58.1% | 59.5% |
| Deeper model | top 1 | 78.1% | 75.4% | 75.7% | 75.6% | 74.4% |
| (Adv. training) | top 5 | 94.1% | 92.6% | 92.7% | 92.5% | 91.6% |

Results of adversarial training using "step l.l." method are provided in Table 1. As it can be seen from the table we were able to significantly increase top-1 and top-5 accuracy on adversarial examples (up to $74\%$ and $92\%$ correspondingly) to make it to be on par with accuracy on clean images. However we lost about $0.8\%$ accuracy on clean examples.

We were able to slightly reduce the gap in the accuracy on clean images by slightly increasing the size of the model. This was done by adding two additional Inception blocks to the model. For specific details about Inception blocks refer to Szegedy et al. (2015).

Unfortunately, training on one-step adversarial examples does not confer robustness to iterative adversarial examples, as shown in Table 2.

Table 2: Accuracy of adversarially trained network on iterative adversarial examples. Adversarial training was done using "step l.l." method. Results were computed after $140k$ iterations of training. Overall, we see that training on one-step adversarial examples does not confer resistance to iterative adversarial examples.

| Adv. method | Training | | Clean | $\epsilon = 2$ | $\epsilon = 4$ | $\epsilon = 8$ | $\epsilon = 16$ |
|---|---|---|---|---|---|---|---|
| Iter. l.l. | Adv. training | top 1 | 77.4% | 29.1% | 7.5% | 3.0% | 1.5% |
| | | top 5 | 93.9% | 56.9% | 21.3% | 9.4% | 5.5% |
| | Baseline | top 1 | 78.3% | 23.3% | 5.5% | 1.8% | 0.7% |
| | | top 5 | 94.1% | 49.3% | 18.8% | 7.8% | 4.4% |
| Iter. basic | Adv. training | top 1 | 77.4% | 30.0% | 25.2% | 23.5% | 23.2% |
| | | top 5 | 93.9% | 44.3% | 33.6% | 28.4% | 26.8% |
| | Baseline | top 1 | 78.3% | 31.4% | 28.1% | 26.4% | 25.9% |
| | | top 5 | 94.1% | 43.1% | 34.8% | 30.2% | 28.8% |

We also tried to use iterative adversarial examples during training, however we were unable to gain any benefits out of it. It is computationally costly and we were not able to obtain robustness to adversarial examples or to prevent the procedure from reducing the accuracy on clean examples significantly. It is possible that much larger models are necessary to achieve robustness to such a large class of inputs.

## 4.2 LABEL LEAKING

We discovered a *label leaking* effect: when a model is trained on FGSM adversarial examples and then evaluated using FGSM adversarial examples, the accuracy on adversarial images becomes much higher than the accuracy on clean images (see Table 3). This effect also occurs (but to a lesser degree) when using other one-step methods that require the true label as input.

We say that label for specific example has been leaked if and only if the model classifies an adversarial example correctly when that adversarial example is generated using the true label but misclassifies a corresponding adversarial example that was created without using the true label. If too many labels has been leaked then accuracy on adversarial examples might become bigger than accuracy on clean examples which we observed on ImageNet dataset.

We believe that the effect occurs because one-step methods that use the true label perform a very simple and predictable transformation that the model can learn to recognize. The adversarial example construction process thus inadvertently leaks information about the true label into the input. We found that the effect vanishes if we use adversarial example construction processes that do not use the true label. The effect also vanishes if an iterative method is used, presumably because the output of an iterative process is more diverse and less predictable than the output of a one-step process.

Overall due to the label leaking effect, we do not recommend to use FGSM or other methods defined with respect to the true class label to evaluate robustness to adversarial examples; we recommend to use other one-step methods that do not directly access the label instead.

We recommend to replace the true label with the most likely label predicted by the model. Alternately, one can maximize the cross-entropy between the full distribution over all predicted labels given the clean input and the distribution over all predicted labels given the perturbed input (Miyato et al., 2016b).

We revisited the adversarially trained MNIST classifier from Goodfellow et al. (2014) and found that it too leaks labels. The most labels are leaked with $\epsilon = 0.3$ on MNIST data in $[0, 1]$. With that $\epsilon$, the model leaks 79 labels on the test set of 10,000 examples. However, the amount of label leaking is small compared to the amount of error caused by adversarial examples. The error rate on adversarial examples exceeds the error rate on clean examples for $\epsilon \in \{.05, .1, .25, .3, .4, .45, .5\}$. This explains why the label leaking effect was not noticed earlier.

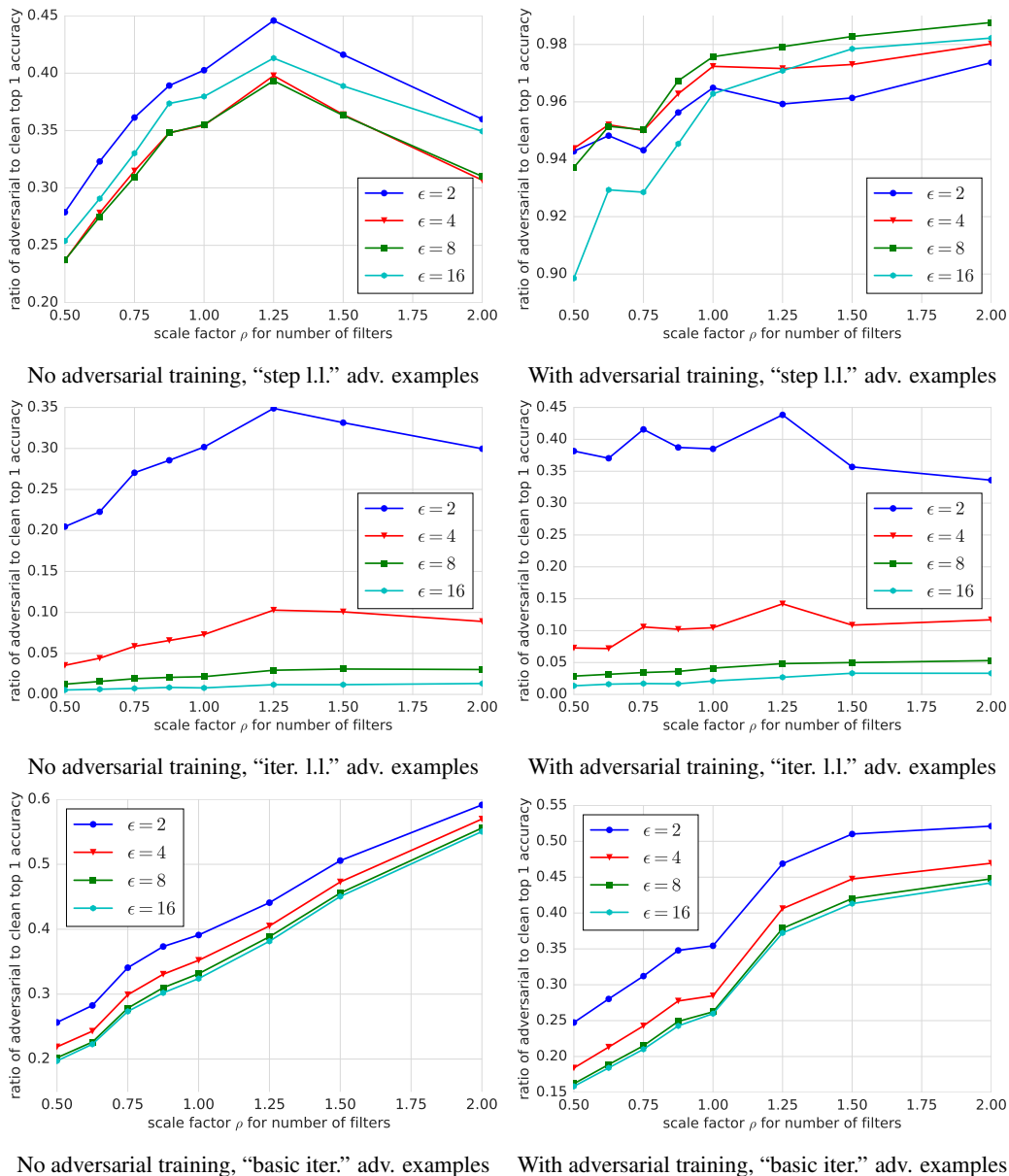

No adversarial training, "step l.l." adv. examples With adversarial training, "step l.l." adv. examples

No adversarial training, "iter. l.l." adv. examples With adversarial training, "iter. l.l." adv. examples

No adversarial training, "basic iter." adv. examples With adversarial training, "basic iter." adv. examples

Figure 1: Influence of size of the model on top 1 classification accuracy of various adversarial examples. Left column — base model without adversarial training, right column — model with adversarial training using "step l.l." method. Top row — results on "step l.l." adversarial images, middle row — results on "iter. l.l." adversarial images, bottom row — results on "basic iter." adversarial images. See text of Section 4.3 for explanation of meaning of horizontal and vertical axes.

Table 3: Effect of label leaking on adversarial examples. When training and evaluation was done using FGSM accuracy on adversarial examples was higher than on clean examples. This effect was not happening when training and evaluation was done using "step l.l." method. In both experiments training was done for $150k$ iterations with initial learning rate $0.0225$.

|  |  | Clean | $\epsilon = 2$ | $\epsilon = 4$ | $\epsilon = 8$ | $\epsilon = 16$ |
|---|---|---|---|---|---|---|
| No label leaking, | top 1 | 77.3% | 72.8% | 73.1% | 73.4% | 72.0% |
| training and eval using "step l.l." | top 5 | 93.7% | 91.1% | 91.1% | 91.0% | 90.3% |
| With label leaking, | top 1 | 76.6% | 86.2% | 87.6% | 88.7% | 87.0% |
| training and eval using FGSM | top 5 | 93.2% | 95.9% | 96.4% | 96.9% | 96.4% |

### 4.3 INFLUENCE OF MODEL CAPACITY ON ADVERSARIAL ROBUSTNESS

We studied how the size of the model (in terms of number of parameters) could affect robustness to adversarial examples. We picked Inception v3 as a base model and varied its size by changing the number of filters in each convolution.

For each experiment we picked a scale factor $\rho$ and multiplied the number of filters in each convolution by $\rho$. In other words $\rho = 1$ means unchanged Inception v3, $\rho = 0.5$ means Inception with half of the usual number of filters in convolutions, etc . . . For each chosen $\rho$ we trained two independent models: one with adversarial training and another without. Then we evaluated accuracy on clean and adversarial examples for both trained models. We have run these experiments for $\rho \in [0.5, 2.0]$.

In earlier experiments (Table 1) we found that *deeper* models benefit more from adversarial training. The increased depth changed many aspects of the model architecture. These experiments varying $\rho$ examine the effect in a more controlled setting, where the architecture remains constant except for the number of feature maps in each layer.

In all experiments we observed that accuracy on clean images kept increasing with increase of $\rho$, though its increase slowed down as $\rho$ became bigger. Thus as a measure of robustness we used the ratio of accuracy on adversarial images to accuracy on clean images because an increase of this ratio means that the gap between accuracy on adversarial and clean images becomes smaller. If this ratio reaches 1 then the accuracy on adversarial images is the same as on clean ones. For a successful adversarial example construction technique, we would never expect this ratio to exceed 1, since this would imply that the adversary is actually helpful. Some defective adversarial example construction techniques, such as those suffering from label leaking, can inadvertently produce a ratio greater than 1.

Results with ratios of accuracy for various adversarial methods and $\epsilon$ are provided in Fig. 1.

For models without adversarial training, we observed that there is an optimal value of $\rho$ yielding best robustness. Models that are too large or too small perform worse. This may indicate that models become more robust to adversarial examples until they become large enough to overfit in some respect.

For adversarially trained models, we found that robustness consistently increases with increases in model size. We were not able to train large enough models to find when this process ends, but we did find that models with twice the normal size have an accuracy ratio approaching 1 for one-step adversarial examples. When evaluated on iterative adversarial examples, the trend toward increasing robustness with increasing size remains but has some exceptions. Also, none of our models was large enough to approach an accuracy ratio of 1 in this regime.

Overall we recommend exploring increase of accuracy (along with adversarial training) as a measure to improve robustness to adversarial examples.

### 4.4 TRANSFERABILITY OF ADVERSARIAL EXAMPLES

From a security perspective, an important property of adversarial examples is that they tend to transfer from one model to another, enabling an attacker in the black-box scenario to create adversarial

Table 4: Transfer rate of adversarial examples generated using different adversarial methods and perturbation size $\epsilon = 16$. This is equivalent to the error rate in an attack scenario where the attacker prefilters their adversarial examples by ensuring that they are misclassified by the source model before deploying them against the target. Transfer rates are rounded to the nearest percent in order to fit the table on the page. The following models were used for comparison: *A* and *B* are Inception v3 models with different random initializations, *C* is Inception v3 model with ELU activations instead of Relu, *D* is Inception v4 model. See also Table 6 for the absolute error rate when the attack is not prefiltered, rather than the transfer rate of adversarial examples.

| | source model | FGSM target model | | | | basic iter. target model | | | | iter l.l. target model | | | |
|---|---|---|---|---|---|---|---|---|---|---|---|---|---|
| | | A | B | C | D | A | B | C | D | A | B | C | D |
| top 1 | A (v3) | 100 | 56 | 58 | 47 | 100 | 46 | 45 | 33 | 100 | 13 | 13 | 9 |
| | B (v3) | 58 | 100 | 59 | 51 | 41 | 100 | 40 | 30 | 15 | 100 | 13 | 10 |
| | C (v3 ELU) | 56 | 58 | 100 | 52 | 44 | 44 | 100 | 32 | 12 | 11 | 100 | 9 |
| | D (v4) | 50 | 54 | 52 | 100 | 35 | 39 | 37 | 100 | 12 | 13 | 13 | 100 |
| top 5 | A (v3) | 100 | 50 | 50 | 36 | 100 | 15 | 17 | 11 | 100 | 8 | 7 | 5 |
| | B (v3) | 51 | 100 | 50 | 37 | 16 | 100 | 14 | 10 | 7 | 100 | 5 | 4 |
| | C (v3 ELU) | 44 | 45 | 100 | 37 | 16 | 18 | 100 | 13 | 6 | 6 | 100 | 4 |
| | D (v4) | 42 | 38 | 46 | 100 | 11 | 15 | 15 | 100 | 6 | 6 | 6 | 100 |

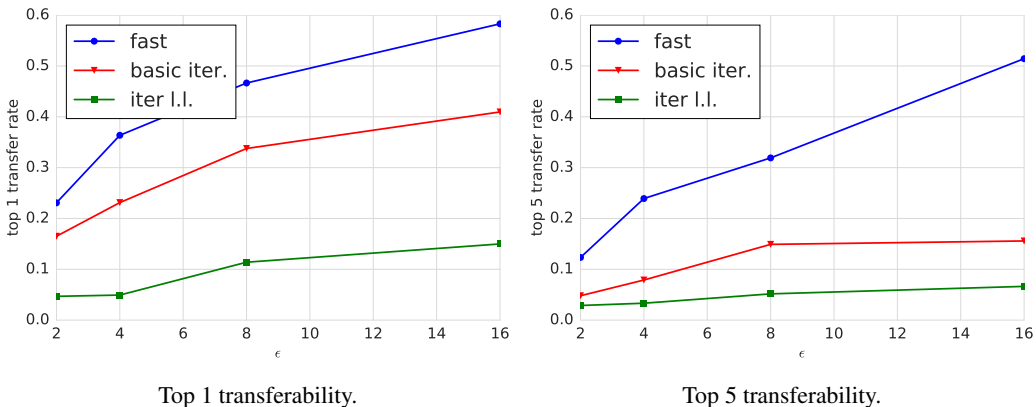

Top 1 transferability.                    Top 5 transferability.

Figure 2: Influence of the size of adversarial perturbation on transfer rate of adversarial examples. Transfer rate was computed using two Inception v3 models with different random intializations. As could be seen from these plots, increase of $\epsilon$ leads to increase of transfer rate. It should be noted that transfer rate is a ratio of number of transferred adversarial examples to number of successful adversarial examples for source network. Both numerator and denominator of this ratio are increasing with increase of $\epsilon$, however we observed that numerator (i.e. number of transferred examples) is increasing much faster compared to increase of denominator. For example when $\epsilon$ increases from 8 to 16 relative increase of denominator is less than 1% for each of the considered methods, at the same time relative increase of numerator is more than 20%.

examples for their own substitute model, then deploy those adversarial examples to fool a target model (Szegedy et al., 2014; Goodfellow et al., 2014; Papernot et al., 2016b).

We studied transferability of adversarial examples between the following models: two copies of normal Inception v3 (with different random initializations and order or training examples), Inception v4 (Szegedy et al., 2016) and Inception v3 which uses ELU activation (Clevert et al., 2015) instead of Relu[2]. All of these models were independently trained from scratch until they achieved maximum accuracy.

---

[2] We achieved 78.0% top 1 and 94.1% top 5 accuracy on Inception v3 with ELU activations, which is comparable with accuracy of Inception v3 model with Relu activations.

In each experiment we fixed the source and target networks, constructed adversarial examples from 1000 randomly sampled clean images from the test set using the source network and performed classification of all of them using both source and target networks. These experiments were done independently for different adversarial methods.

We measured transferability using the following criteria. Among 1000 images we picked only misclassified adversarial example for the source model (i.e. clean classified correctly, adversarial misclassified) and measured what fraction of them were misclassified by the target model.

Transferability results for all combinations of models and $\epsilon = 16$ are provided in Table 4. Results for various $\epsilon$ but fixed source and target model are provided in Fig. 2.

As can be seen from the results, FGSM adversarial examples are the most transferable, while "iter l.l." are the least. On the other hand "iter l.l." method is able to fool the network in more than 99% cases (top 1 accuracy), while FGSM is the least likely to fool the network. This suggests that there might be an inverse relationship between transferability of specific method and ability of the method to fool the network. We haven't studied this phenomenon further, but one possible explanation could be the fact that iterative methods tend to overfit to specific network parameters.

In addition, we observed that for each of the considered methods transfer rate is increasing with increase of $\epsilon$ (see Fig. 2). Thus potential adversary performing a black-box attack have an incentive to use higher $\epsilon$ to increase the chance of success of the attack.

## 5 CONCLUSION

In this paper we studied how to increase robustness to adversarial examples of large models (Inception v3) trained on large dataset (ImageNet). We showed that adversarial training provides robustness to adversarial examples generated using one-step methods. While adversarial training didn't help much against iterative methods we observed that adversarial examples generated by iterative methods are less likely to be transferred between networks, which provides indirect robustness against black box adversarial attacks. In addition we observed that increase of model capacity could also help to increase robustness to adversarial examples especially when used in conjunction with adversarial training. Finally we discovered the effect of label leaking which resulted in higher accuracy on FGSM adversarial examples compared to clean examples when the network was adversarially trained.

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

# Appendices

## A   COMPARISON OF ONE-STEP ADVERSARIAL METHODS

In addition to FGSM and "step l.l." methods we explored several other one-step adversarial methods both for training and evaluation. Generally all of these methods can be separated into two large categories. Methods which try to maximize the loss (similar to FGSM) are in the first category. The second category contains methods which try to maximize the probability of a specific target class (similar to "step l.l."). We also tried to use different types of random noise instead of adversarial images, but random noise didn't help with robustness against adversarial examples.

The full list of one-step methods we tried is as follows:

- Methods increasing loss function $J$
  - FGSM (described in details in Section 2.2):
  $$\boldsymbol{X}^{adv} = \boldsymbol{X} + \epsilon \operatorname{sign}\big(\nabla_X J(\boldsymbol{X}, y_{true})\big)$$
  - FGSM-pred or fast method with predicted class. It is similar to FGSM but uses the label of the class predicted by the network instead of true class $y_{true}$.
  - "Fast entropy" or fast method designed to maximize the entropy of the predicted distribution, thereby causing the model to become less certain of the predicted class.
  - "Fast grad. $L_2$" is similar to FGSM but uses the value of gradient instead of its sign. The value of gradient is normalized to have unit $L_2$ norm:
  $$\boldsymbol{X}^{adv} = \boldsymbol{X} + \epsilon \frac{\nabla_X J(\boldsymbol{X}, y_{true})}{\big\|\nabla_X J(\boldsymbol{X}, y_{true})\big\|_2}$$
  Miyato et al. (2016b) advocate this method.
  - "Fast grad. $L_\infty$" is similar to "fast grad. $L_2$" but uses $L_\infty$ norm for normalization.
- Methods increasing the probability of the selected target class
  - "Step l.l." is one-step towards least likely class (also described in Section 2.2):
  $$\boldsymbol{X}^{adv} = \boldsymbol{X} - \epsilon \operatorname{sign}\big(\nabla_X J(\boldsymbol{X}, y_{target})\big)$$
  where $y_{target} = \arg\min_y \{p(y \mid \boldsymbol{X})\}$ is least likely class prediction by the network.
  - "Step rnd." is similar to "step l.l." but uses random class instead of least likely class.
- Random perturbations
  - Sign of random perturbation. This is an attempt to construct random perturbation which has similar structure to perturbations generated by FGSM:
  $$\boldsymbol{X}^{adv} = \boldsymbol{X} + \epsilon \operatorname{sign}\big(\mathcal{N}\big)$$
  where $\mathcal{N}$ is random normal variable with zero mean and identity covariance matrix.
  - Random truncated normal perturbation with zero mean and $0.5\epsilon$ standard deviation defined on $[-\epsilon, \epsilon]$ and uncorrelated pixels, which leads to the following formula for perturbed images:
  $$\boldsymbol{X}^{adv} = \boldsymbol{X} + \mathcal{T}$$
  where $\mathcal{T}$ is a random variable with truncated normal distribution.

Overall, we observed that using only one of these single step methods during adversarial training is sufficient to gain robustness to all of them. Fig. 3 shows accuracy on various one-step adversarial examples when the network was trained using only "step l.l." method.

At the same time we observed that not all one-step methods are equally good for adversarial training, as shown in Table 5. The best results (achieving both good accuracy on clean data and good accuracy on adversarial inputs) were obtained when adversarial training was done using "step l.l." or "step rnd." methods.

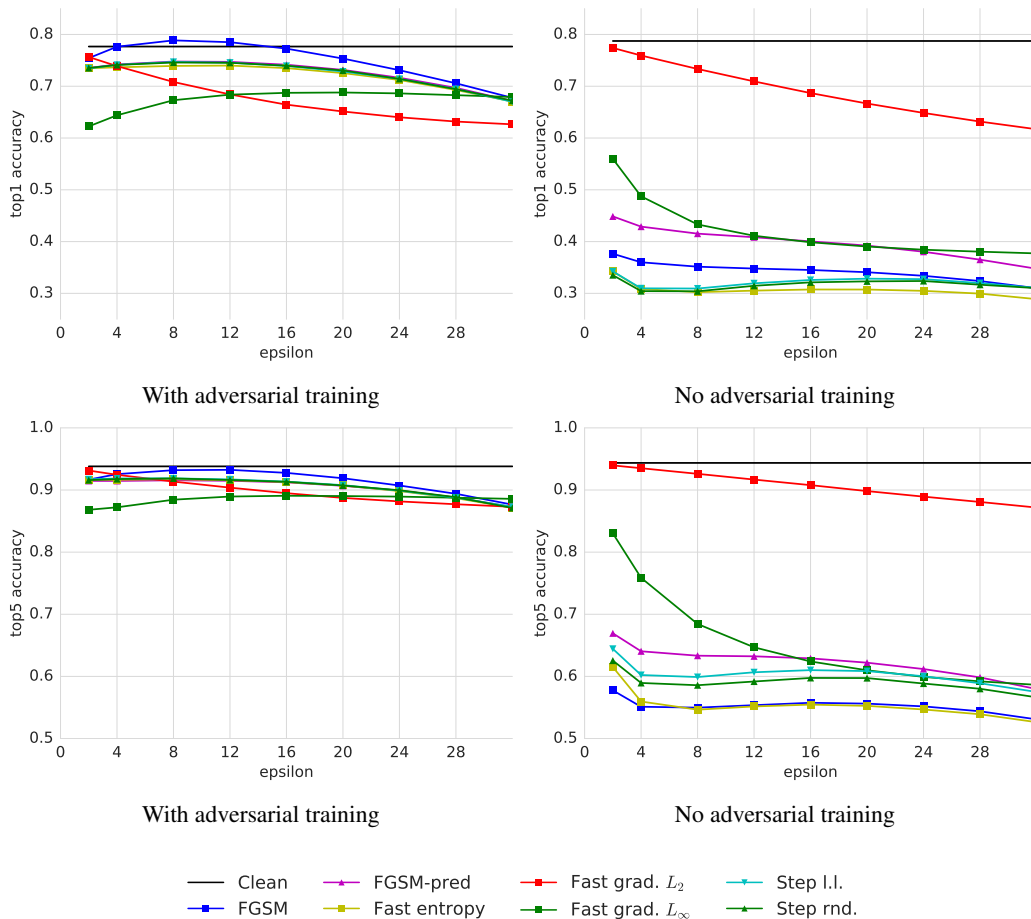

Figure 3: Comparison of different one-step adversarial methods during eval. Adversarial training was done using "step l.l." method. Some evaluation methods show increasing accuracy with increasing $\epsilon$ over part of the curve, due to the label leaking effect.

## B    ADDITIONAL RESULTS WITH SIZE OF THE MODEL

Section 4.3 contains details regarding the influence of size of the model on robustness to adversarial examples. Here we provide additional Figure 4 which shows robustness calculated using top 5 accuracy. Generally it exhibits the same properties as the corresponding plots for top 1 accuracy.

## C    ADDITIONAL RESULTS ON TRANSFERABILITY

Section 4.4 contains results with transfer rate of various adversarial examples between models. In addition to transfer rate computed only on misclassified adversarial examples it is also interesting to observe the error rate of all candidate adversarial examples generated for one model and classified by other model.

This result might be interesting because it models the following attack. Instead of trying to pick "good" adversarial images an adversary tries to modify all available images in order to get as much misclassified images as possible.

To compute the error rate we randomly generated 1000 adversarial images using the source model and then classified them using the target model. Results for various models, adversarial methods

Table 5: Comparison of different one-step adversarial methods for adversarial training. The evaluation was run after $90k$ training steps.
*) In all cases except "fast grad $L_2$" and "fast grad $L_\infty$" the evaluation was done using FGSM. For "fast grad $L_2$" and "fast grad $L_\infty$" the evaluation was done using "step l.l." method. In the case where both training and testing were done with FGSM, the performance on adversarial examples is artificially high due to the label leaking effect. Based on this table, we recommend using "step rnd." or "step l.l." as the method of generating adversarial examples at training time, in order to obtain good accuracy on both clean and adversarial examples. We computed 95% confidence intervals based on the standard error of the mean around the test error, using the fact that the test error was evaluated with 50,000 samples. Within each column, we indicate which methods are statistically tied for the best using bold face.

|  | Clean | $\epsilon = 2$ | $\epsilon = 4$ | $\epsilon = 8$ | $\epsilon = 16$ |
|---|---|---|---|---|---|
| No adversarial training | **76.8%** | 40.7% | 39.0% | 37.9% | 36.7% |
| FGSM | 74.9% | ~~79.3%~~ | ~~82.8%~~ | ~~85.3%~~ | ~~83.2%~~ |
| Fast with predicted class | **76.4%** | 43.2% | 42.0% | 40.9% | 40.0% |
| Fast entropy | **76.4%** | 62.8% | 61.7% | 59.5% | 54.8% |
| Step rnd. | **76.4%** | **73.0%** | **75.4%** | **76.5%** | **72.5%** |
| Step l.l. | **76.3%** | **72.9%** | **75.1%** | **76.2%** | **72.2%** |
| Fast grad. $L_2$* | **76.8%** | 44.0% | 33.2% | 26.4% | 22.5% |
| Fast grad. $L_\infty$* | 75.6% | 52.2% | 39.7% | 30.9% | 25.0% |
| Sign of random perturbation | **76.5%** | 38.8% | 36.6% | 35.0% | 32.7% |
| Random normal perturbation | **76.6%** | 38.3% | 36.0% | 34.4% | 31.8% |

and fixed $\epsilon = 16$ are provided in Table 6. Results for fixed source and target models and various $\epsilon$ are provided in Fig. 5.

Overall the error rate of transferred adversarial examples exhibits the same behavior as the transfer rate described in Section 4.4.

Table 6: Error rates on adversarial examples transferred between models, rounded to the nearest percent. Results are provided for adversarial images generated using different adversarial methods and fixed perturbation size $\epsilon = 16$. The following models were used for comparison: *A* and *B* are Inception v3 models with different random initializations, *C* is Inception v3 model with ELU activations instead of Relu, *D* is Inception v4 model. See also Table 4 for the transfer rate of adversarial examples, rather than the absolute error rate.

|  |  | FGSM | | | | basic iter. | | | | iter l.l. | | | |
|---|---|---|---|---|---|---|---|---|---|---|---|---|---|
|  | source | target model | | | | target model | | | | target model | | | |
|  | model | A | B | C | D | A | B | C | D | A | B | C | D |
| top 1 | A (v3) | 65 | 52 | 53 | 45 | 78 | 51 | 50 | 42 | 100 | 32 | 31 | 27 |
|  | B (v3) | 52 | 66 | 54 | 48 | 50 | 79 | 51 | 43 | 35 | 99 | 34 | 29 |
|  | C (v3 ELU) | 53 | 55 | 70 | 50 | 47 | 46 | 74 | 40 | 31 | 30 | 100 | 28 |
|  | D (v4) | 47 | 51 | 49 | 62 | 43 | 46 | 45 | 73 | 30 | 31 | 31 | 99 |
| top 5 | A (v3) | 46 | 28 | 28 | 22 | 76 | 17 | 18 | 13 | 94 | 12 | 12 | 9 |
|  | B (v3) | 29 | 46 | 30 | 22 | 19 | 76 | 18 | 16 | 13 | 96 | 12 | 11 |
|  | C (v3 ELU) | 28 | 29 | 55 | 25 | 18 | 19 | 74 | 15 | 12 | 12 | 96 | 9 |
|  | D (v4) | 23 | 22 | 25 | 40 | 14 | 16 | 16 | 70 | 11 | 11 | 11 | 97 |

# D   RESULTS WITH DIFFERENT ACTIVATION FUNCTIONS

We evaluated robustness to adversarial examples when the network was trained using various non-linear activation functions instead of the standard $relu$ activation when used with adversarial training on "step l.l." adversarial images. We tried to use following activation functions instead of $relu$:

- $tanh(x)$

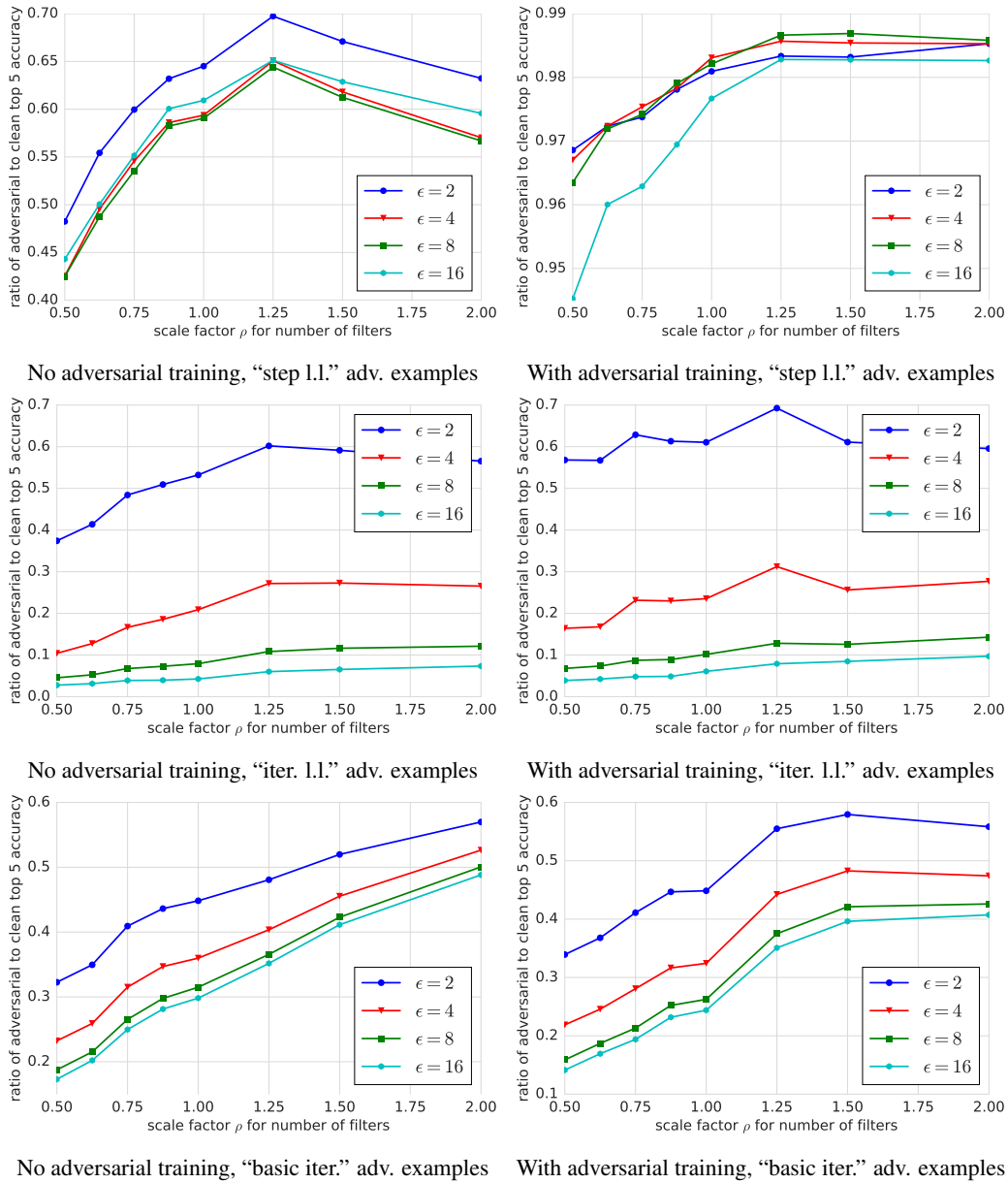

No adversarial training, "step l.l." adv. examples    With adversarial training, "step l.l." adv. examples

No adversarial training, "iter. l.l." adv. examples    With adversarial training, "iter. l.l." adv. examples

No adversarial training, "basic iter." adv. examples    With adversarial training, "basic iter." adv. examples

Figure 4: Influence of size of the model on top 5 classification accuracy of various adversarial examples. For a detailed explanation see Section 4.3 and Figure 1.

- $relu6(x) = min(relu(x), 6)$

- $ReluDecay_\beta(x) = \frac{relu(x)}{1 + \beta relu(x)^2}$ for $\beta \in \{0.1, 0.01, 0.001\}$

Training converged using all of these activations, however test performance was not necessarily the same as with $relu$.

$tanh$ and $ReluDecay_{\beta=0.1}$ lose about 2%-3% of accuracy on clean examples and about 10%-20% on "step l.l." adversarial examples. $relu6$, $ReluDecay_{\beta=0.01}$ and $ReluDecay_{\beta=0.001}$ demonstrated similar accuracy (within ±1%) to $relu$ on clean images and few percent loss of accuracy on "step l.l." images. At the same time all non-linear activation functions increased classification accuracy on some of the iterative adversarial images. Detailed results are provided in Table 7.

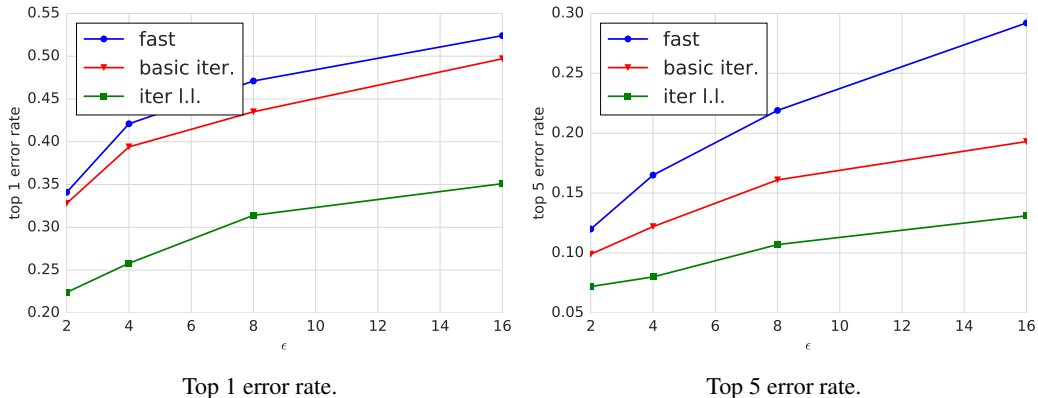

Top 1 error rate.                    Top 5 error rate.

Figure 5: Influence of the size of adversarial perturbation on the error rate on adversarial examples generated for one model and classified using another model. Both source and target models were Inception v3 networks with different random intializations.

Overall non linear activation functions could be used as an additional measure of defense against iterative adversarial images.

Table 7: Activation functions and robustness to adversarial examples. For each activation function we adversarially trained the network on "step l.l." adversarial images and then run classification of clean images and adversarial images generated using various adversarial methods and $\epsilon$.

| Adv. method | Activation | Clean | $\epsilon = 2$ | $\epsilon = 4$ | $\epsilon = 8$ | $\epsilon = 16$ |
|---|---|---|---|---|---|---|
| Step l.l. | $relu$ | 77.5% | 74.6% | 75.1% | 75.5% | 74.5% |
| | $relu6$ | 77.7% | 71.8% | 73.5% | 74.5% | 74.0% |
| | $ReluDecay_{0.001}$ | 78.0% | 74.0% | 74.9% | 75.2% | 73.9% |
| | $ReluDecay_{0.01}$ | 77.4% | 73.6% | 74.6% | 75.0% | 73.6% |
| | $ReluDecay_{0.1}$ | 75.3% | 67.5% | 67.5% | 67.0% | 64.8% |
| | $tanh$ | 74.5% | 63.7% | 65.1% | 65.8% | 61.9% |
| Iter. l.l. | $relu$ | 77.5% | 30.2% | 8.0% | 3.1% | 1.6% |
| | $relu6$ | 77.7% | 39.8% | 13.7% | 4.1% | 1.9% |
| | $ReluDecay_{0.001}$ | 78.0% | 39.9% | 12.6% | 3.8% | 1.8% |
| | $ReluDecay_{0.01}$ | 77.4% | 36.2% | 11.2% | 3.2% | 1.6% |
| | $ReluDecay_{0.1}$ | 75.3% | 47.0% | 25.8% | 6.5% | 2.4% |
| | $tanh$ | 74.5% | 35.8% | 6.6% | 2.7% | 0.9% |
| Basic iter. | $relu$ | 77.5% | 28.4% | 23.2% | 21.5% | 21.0% |
| | $relu6$ | 77.7% | 31.2% | 26.1% | 23.8% | 23.2% |
| | $ReluDecay_{0.001}$ | 78.0% | 32.9% | 27.2% | 24.7% | 24.1% |
| | $ReluDecay_{0.01}$ | 77.4% | 30.0% | 24.2% | 21.4% | 20.5% |
| | $ReluDecay_{0.1}$ | 75.3% | 26.7% | 20.6% | 16.5% | 15.2% |
| | $tanh$ | 74.5% | 24.5% | 22.0% | 20.9% | 20.7% |

# E  RESULTS WITH DIFFERENT NUMBER OF ADVERSARIAL EXAMPLES IN THE MINIBATCH

We studied how number of adversarial examples $k$ in the minibatch affect accuracy on clean and adversarial examples. Results are summarized in Table 8.

Overall we noticed that increase of $k$ lead to increase of accuracy on adversarial examples and to decrease of accuracy on clean examples. At the same having more than half of adversarial examples in the minibatch (which correspond to $k > 16$ in our case) does not provide significant improvement of accuracy on adversarial images, however lead to up to $1\%$ of additional decrease of accuracy

on clean images. Thus for most experiments in the paper we have chosen $k = 16$ as a reasonable trade-off between accuracy on clean and adversarial images.

Table 8: Results of adversarial training depending on $k$ — number of adversarial examples in the minibatch. Adversarial examples for training and evaluation were generated using step l.l. method. Row 'No adv' is a baseline result without adversarial training (which is equivalent to $k = 0$). Rows 'Adv, $k = X$' are results of adversarial training with $X$ adversarial examples in the minibatch. Total minibatch size is 32, thus $k = 32$ correspond to minibatch without clean examples.

|  | Clean | $\epsilon = 2$ | $\epsilon = 4$ | $\epsilon = 8$ | $\epsilon = 16$ |
|---|---|---|---|---|---|
| No adv | 78.2% | 31.5% | 27.7% | 27.8% | 29.7% |
| Adv, $k = 4$ | 78.3% | 71.7% | 71.3% | 69.4% | 65.8% |
| Adv, $k = 8$ | 78.1% | 73.2% | 73.2% | 72.6% | 70.5% |
| Adv, $k = 16$ | 77.6% | 73.8% | 75.3% | 76.1% | 75.4% |
| Adv, $k = 24$ | 77.1% | 73.0% | 75.3% | 76.2% | 76.0% |
| Adv, $k = 32$ | 76.3% | 73.4% | 75.1% | 75.9% | 75.8% |

