# Peer review of "Adversarial Machine Learning at Scale"

_ICLR 2017 — accepted_

[Official Review · AnonReviewer3 · rating 6 · confidence 4 · 18 Dec 2016]
**Official review**

This paper investigate the phenomenon of the adversarial examples and the adversarial training on the dataset of ImageNet. While the final conclusions are still vague, this paper raises several noteworthy finding from its experiments.
The paper is well written and easy to follow. Although I still have some concerns about the paper (see the comments below), this paper has good contributions and worth to publish.

Pros:
For the first time in the literature, this paper proposed the concept of ‘label leaking’. Although its effect only becomes significant when the dataset is large, it should be carefully handled in the future research works along this line.
Using the ratio of 'clean accuracy' over ‘adversarial accuracy’ as the measure of robust is more reasonable compared to the existing works in the literature. 

Cons:
Although the conclusions of the paper are based on the experiments on ImageNet, the title of the paper seems a little misleading. I consider Section 4 as the main contribution of the paper. Note that Section 4.3 and Section 4.4 are not specific to large-scale dataset, thus emphasizing the ‘large-scale’ in the title and in the introduction seems improper. 
Basically all the conclusions of the paper are made based on observing the experimental results. Further tests should have been performed to verify these hypotheses. Without that, the conclusions of the paper seems rushy. For example, one dataset of imageNet can not infer the conclusions for all large-scale datasets.

[Official Review · AnonReviewer1 · rating 6 · confidence 4 · 18 Dec 2016]
**No Title**

This paper is a well written paper. This paper can be divided into 2 parts:
1.Adversary training on ImageNet 
2.Empirical study of label leak, single/multiple step attack, transferability and importance of model capacity

For part [1], I don’t think training without clean example will not make reasonable ImageNet level model. Ian’s experiment in “Explaining and Harnessing Adversarial Examples” didn't use BatchNorm, which may be important for training large scale model. This part looks like an extension to Ian’s work with Inception-V3 model. I suggest to add an experiment of training without clean samples.

For part [2], The experiments cover most variables in adversary training, yet lack technical depth.  The depth, model capacity experiments can be explained by regularizer effect of adv training;  Label leaking is novel; In transferability experiment with FGSM, if we do careful observe on some special MNIST FGSM example, we can find augmentation effect on numbers, which makes grey part on image to make the number look more like the other numbers. Although this effect is hard to be observed with complex data such as CIFAR-10 or ImageNet, they may be related to the authors' observation "FGSM examples are most transferable".  

In this part the authors raise many interesting problems or guess, but lack theoretical explanations. 

Overall I think these empirical observations are useful for future work.

[Official Review · AnonReviewer2 · rating 7 · confidence 3 · 19 Dec 2016]

This paper has two main contributions: 
(1) Applying adversarial training to imagenet, a larger dataset than previously considered 
(2) Comparing different adversarial training approaches, focusing importantly on the transferability of different methods. The authors also uncover and explain the label leaking effect which is an important contribution.

This paper is clear, well written and does a good job of assessing and comparing adversarial training methods and understanding their relation to one another. A wide range of empirical results are shown which helps elucidate the adversarial training procedure. This paper makes an important contribution towards understand adversarial training and believe ICLR is an appropriate venue for this work.

[Final Decision · Program Chairs · 06 Feb 2017]
**ICLR committee final decision**

This paper is an empirical study of adversarial training in a large-scale regime. 
 Its main contributions are to demonstrate that Imagenet models can be made robust to adversarial examples using relatively efficient constructions (so-called 'one-step' methods). Along the way, the authors also report the 'label leaking' effect, a flaw in previous adversarial constructions that limits the effectiveness of adversarial examples at regularizing training. 
 
 The reviewers were consensual that this contribution is useful to the field, and the label leaking phenomena is an important aspect that this paper helps address and mitigate. The authors responded promptly to reviewers questions. Despite limited novelty and lack of quantitative analysis, I recommend accept as a poster.